# Holographic colour prints for enhanced optical security by combined phase and amplitude control

Kevin T.P. Lim [1,3], Hailong Liu[1], Yejing Liu [1] & Joel K.W. Yang [1,2]

Conventional optical security devices provide authentication by manipulating a specific property of light to produce a distinctive optical signature. For instance, microscopic colour prints modulate the amplitude, whereas holograms typically modulate the phase of light. However, their relatively simple structure and behaviour is easily imitated. We designed a pixel that overlays a structural colour element onto a phase plate to control both the phase and amplitude of light, and arrayed these pixels into monolithic prints that exhibit complex behaviour. Our fabricated prints appear as colour images under white light, while projecting up to three different holograms under red, green, or blue laser illumination. These holographic colour prints are readily verified but challenging to emulate, and can provide enhanced security in anti-counterfeiting applications. As the prints encode information only in the surface relief of a single polymeric material, nanoscale 3D printing of customised masters may enable their mass-manufacture by nanoimprint lithography.

[1] Singapore University of Technology and Design (SUTD), 8 Somapah Road, Singapore 487372, Singapore. [2] Institute of Materials Research and Engineering (IMRE), 2 Fusionopolis Way, Innovis, #08-03, Singapore 138634, Singapore. [3]Present address: Cavendish Laboratory, JJ Thomson Ave, Cambridge CB3 0HE, United Kingdom. These authors contributed equally: Kevin T. P. Lim, Hailong Liu. Correspondence and requests for materials should be addressed to K.T.P.L. (email: tpkl2@cam.ac.uk) or to J.K.W.Y. (email: joel_yang@sutd.edu.sg)

Optical security devices are valuable tools in data encryption and document authentication as they exploit the many properties of light, including amplitude, phase, polarisation, and wavelength, to create distinctive visual effects that can be difficult to decode or duplicate[1,2]. The two archetypal optical security devices are microscopic colour prints[3–5] and holograms[6]. Microscopic colour images can be directly viewed under a magnifying glass, whereas holograms are easily verified by using a laser pointer to project an image onto a screen placed in the far field (Fraunhofer regime). To strengthen the security of these basic devices, additional complexity can be introduced by encoding multiple sets of information into a single device, i.e. multiplexing.

Multiplexed colour prints have been created by encoding information in two independent dimensions of elongated metal nanostructures, allowing for two different images to be read out under orthogonal polarisations of light[7,8]. Using similar nanostructures of various sizes optimised to respond at different wavelengths, three-colour multiplexed holograms based on the Pancharatnam–Berry (PB) geometric phase have also been demonstrated[9–13]. Unfortunately, transmission PB holograms often suffer from low transmission efficiency due to their use of lossy metal nanostructures and are complicated to read out, requiring the use of circularly polarised light as well as specific illumination and/or viewing angles. Additionally, the arrays of nanostructures are fabricated with electron beam lithography or focused ion beam milling, which incurs high costs and imposes practical constraints on the patternable area. These shortcomings, namely low transmission efficiency, complexity of readout, and high fabrication costs, have limited the practical application of metal nanostructure-based colour prints and holograms in optical security devices thus far.

In comparison, traditional phase elements consisting of dielectric structures of different thicknesses (phase plates) enable holographic projection to be achieved with higher transmission efficiency, simpler illumination methods (e.g. a handheld laser pointer), and little restriction on the polarisation or incidence angle of the light. They are also usually easier and cheaper to manufacture than metal nanostructures as their larger dimensions are within the resolution limit of photolithography. High transmission efficiency multiplexed holograms that project up to three different images depending on the incident wavelength have previously been demonstrated using a variety of techniques including phase modulation[14–16] and depth division[17]. Recently, colour holograms[18] operating in transmission under white light have also been developed.

However, as phase holograms are not designed to control the amplitude of light, they generally appear random or featureless under incoherent illumination, which makes them less attractive as optical security devices. Conversely, colour images have superior decorative values on banknotes or documents but generally cannot produce any meaningful holographic projection under coherent illumination as they do not control the phase of the light. Introducing a design methodology to control the phase and amplitude of light simultaneously is an area that has been relatively unexplored and could enable the creation of a dual-function device that appears as an image in plain view, but encrypts additional data that can be retrieved through holographic projection. By encryption we refer to the fact that the holographic information cannot be read except by illumination with coherent monochromatic light of a suitable wavelength, noting that the introduction of one or more additional phase masks as security keys[19] could in principle enable even more secure encryption.

Here, we propose an optical security device that combines phase and amplitude control to integrate (multiple) holograms into a colour print, which we refer to as a (multiplexed) holographic colour print. Our device appears as a colour image when viewed in white light, but reveals up to three different hidden grayscale holographic projections under red, green, and blue laser illumination. To the best of our knowledge, this is the first time that holograms have been encrypted into a colour print, which we achieve by encoding both phase and colour independently within individual pixel elements. Our holographic colour prints provide a unique and easily recognisable visual effect, and may be of interest to the security industry as effective anti-counterfeiting elements that provide enhanced optical security on important documents such as identity cards and passports.

## Results

**Concept of a holographic colour print**. The concept of a transmission holographic colour print is illustrated in Fig. 1. The top layer contains colour filters that encode a colour print, and the bottom layer contains phase plates that encode the holograms. The colour filters have two functions: (1) to collectively form a colour image under white light illumination, and (2) to control the transmission of red, green, and blue (RGB) laser light through the pixels of the underlying multiplexed holograms. With coherent monochromatic illumination (e.g. light from a laser), the incident light is filtered such that only the relevant phase plates with colour filters that closely match the illumination wavelength are selected for a given holographic projection, whereas the other phase plates do not form a projection as their colour filters are mismatched and reject the incident light. Therefore, the multiplexed holographic colour print will show different holographic projections when illuminated by red, green, and blue lasers. Because incident light passes through all pixels in parallel, the pixels can act independently to allow transmission of different wavelengths in some regions of space but not others, which enables several holograms to jointly occupy the total area available in a spatial multiplexing scheme. Using the freedom to divide the space into regions of arbitrary shapes and sizes, the individual hologram areas can then be strategically allocated such that the arrangement of their colour filters additionally encodes a chosen colour image. Under incoherent white light illumination (e.g. light from a lamp or torch), the phase modulation of the holograms is effectively ignored and the colour filters act as amplitude-modulating colour pixels that together show the desired colour image.

**Design of a holographic colour pixel**. To create a physical realisation of a holographic colour print, we first developed a holographic colour pixel that controls both the phase and amplitude of light. As our pixel has a relatively large minimum feature size of several hundreds of nanometres and consists entirely of a single dielectric material, we are able to fabricate it with femtosecond 3D printing (direct laser writing) as a monolithic structure in a cross-linked polymer (see Methods section for details of the fabrication process used).

Our pixel design (Fig. 2a) integrates a dielectric phase plate under a structural colour element comprising an array of dielectric pillars, which acts as a colour pixel for the transmission colour image under white light illumination and also as a colour filter[20] to selectively transmit red, green, or blue laser light for hologram multiplexing. The colour filters are diffractive in nature, transmitting the desired wavelengths of light on-axis and rejecting unwanted wavelengths by diffracting them off-axis at large angles (Supplementary Fig. 5). The dielectric phase plate controls the phase of transmitted light according to the equation $\phi(\lambda) = 2\pi(n-1)t/\lambda$, where the phase shift ($\phi$) arises from path length differences that depend on the phase plate thickness ($t$) and refractive index ($n$). The refractive index of the dielectric polymer material we used (between 1.54 and 1.58 in the visible region)[21] allows us to achieve a full $2\pi$ phase modulation for red, green, and blue light by varying the phase plate thickness over a range of 1.2 μm. As it is impractical for us to fabricate samples

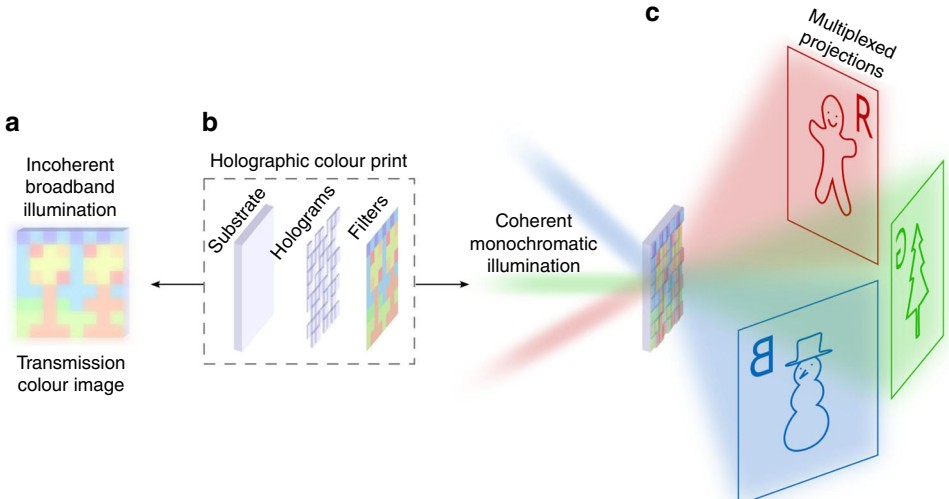

**Fig. 1** Illustration of a holographic colour print. Exploded-view schematic of a transmission holographic colour print (**b**), a layered optical device in which colour filters are integrated on top of holograms. The colour filters act as colour pixels in a colour image under white light illumination from a lamp or torch (**a**) and also serve to control the transmission of red, green, and blue (RGB) laser light through the pixels of the underlying multiplexed holograms (**c**). Under RGB laser illumination, each wavelength of light selects a different holographic projection, which is independent of the colour image and the other projections. Far field projections appear along the axis of laser illumination. Three different angles of incidence are shown to illustrate the three distinct holographic projections. The projections remain in focus at any distance in the far field, can be achieved over a wide range of incident angles, and are perfectly overlapped under collinear multi-colour illumination

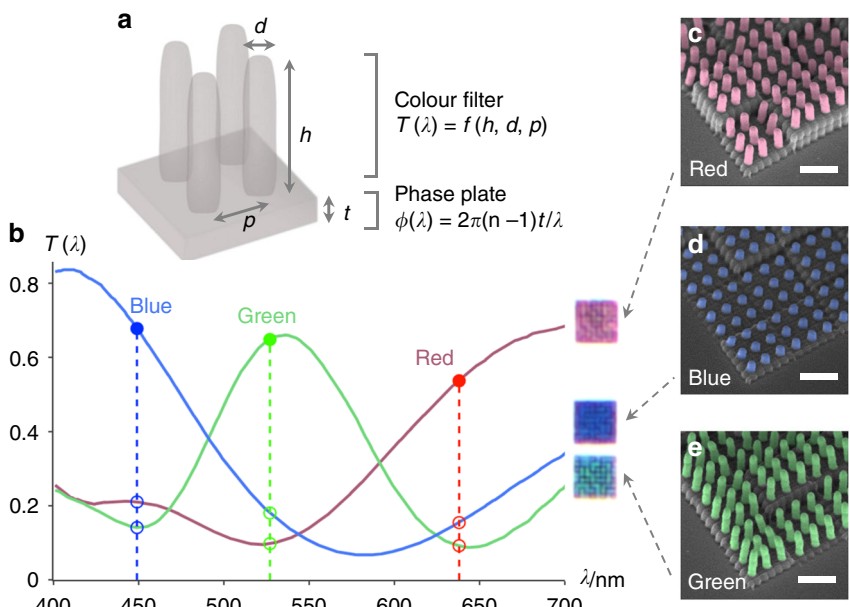

**Fig. 2** Structure and characteristics of a holographic colour pixel. **a** Schematic of the structure of a holographic colour pixel that provides combined phase and amplitude control, comprising an array of pillars (colour filter) integrated on top of a block (phase plate) made of a dielectric material with refractive index $n$. The colour filter controls the amplitude of light based on its transmission spectrum $T(\lambda) = f(h, d, p)$, which depends on the pillar array dimensions $\{h, d, p\}$ (height, diameter, and pitch). The phase plate controls the phase of transmitted light where the phase shift arises from path length differences that depend on the block thickness. **b** Transmission spectra and corresponding optical micrographs of pillar arrays that serve as red, green and blue colour filters. Transmission spectra were averaged from measurements of pillar arrays with the same dimensions as those shown in the images, but patterned separately on blocks of uniform thickness (0.6, 1.0, 1.4, 1.8 μm). Good RGB wavelength selectivity can be seen from the high transmittance (average 62%) for red (638 nm), green (527 nm), and blue (449 nm) light passing through matching colour filters (filled circles), and low transmittance (average 15%) for light passing through mismatched filters (empty circles). **c**–**e** False-colour tilt-view SEMs of pillar arrays with dimensions optimised to give the best selectivity for red, green, and blue light. The pillars (~0.4 μm in diameter and respectively 1.9, 0.7, and 2.6 μm in height) are patterned in a square array of 1 μm pitch onto $3 \times 3\ \mu m^2$ blocks of randomly varying thickness within the thickness range to be used for hologram phase plates (0.6–1.8 μm). Scale bars are 2 μm

with a continuous phase variation, we quantise the phase and define a number of discrete thickness steps to span the required 1.2 μm range (Supplementary Methods).

Assuming that a phase plate acts as an ideal phase-controlling (constant-amplitude) element and a pillar array colour filter acts as an ideal amplitude-controlling (constant-phase) element, combining these elements into a layered pixel should allow for independent phase and amplitude control. In practice, however, due to the refractive index mismatch between the glass substrate and the polymer structures, changing the thickness of the underlying block to control the phase also affected the transmission amplitude of the overall pixel, i.e. phase-amplitude coupling was present. The shift in transmission spectrum with block thickness caused a significant change in the pixel colour with thin blocks of $t = 0–0.4$ μm, but was not noticeable for thicker blocks of $t \geq 0.6$ μm (Supplementary Fig. 1). As such, we chose to work with blocks of 0.6–1.8 μm thickness, covering the required range of 1.2 μm.

To minimise any remaining variations in pixel transmission amplitude due to differences in phase plate thickness, we fabricated dielectric pillar arrays on blocks with thicknesses varying between 0.6 to 1.8 μm and measured their transmission spectra $T(\lambda)$. We could then average out the dependence of $T(\lambda)$ on thickness, effectively eliminating any residual thickness dependence. Subsequently, varying the pillar array dimensions of height ($h$), diameter ($d$), and pitch ($p$) enabled us to access a range of colours spanning > 50% of the sRGB colour gamut (Supplementary Fig. 2), from which we selected the most suitable filters for red, green, and blue wavelengths (Supplementary Note 1). The transmission spectra of the RGB colour filters (Fig. 2b) with optimised pillar array dimensions (Fig. 2c–e) show a high transmittance averaging 62% at the desired wavelength and a low transmittance averaging 15% at unwanted wavelengths (Supplementary Table 1). These transmittance values afford sufficient wavelength selectivity, i.e. mutually exclusive or orthogonal transmission, at the wavelengths of interest (Supplementary Table 2). The wide colour range enables us to choose suitable colours to reproduce the colour image under white light illumination, and the good wavelength selectivity ensures that laser light can be filtered to distinguish the individual holograms.

In addition to reducing the effects of pixel phase on pixel amplitude (phase-amplitude coupling), which would otherwise affect multiplexing and colour image formation, we also investigated the effect of pixel amplitude on pixel phase (amplitude-phase coupling), which could affect the holographic projections. Indeed, the pillar colour filters add a weak unwanted phase variation on top of the desired phase variation controlled by the phase plate thickness (Supplementary Figs. 8 and 9). While this extra phase can be compensated for at the pixel level if necessary, we found that it was an order of magnitude smaller than the effect of the phase plates (Supplementary Note 4) and could be safely neglected when designing our holographic colour prints.

**Fabrication of holographic colour prints**. With our holographic colour pixel design, it is possible to create multiplexed holograms by fabricating large arrays of pixels. In the simplest case, holograms can be multiplexed with the phase plates of each hologram spatially segregated in contiguous single-coloured regions, giving a similar result to that obtained by attaching macroscopic colour filters side by side on a spatial light modulator[22]. However, this multiplexing scheme cannot be used to realise an arbitrary colour image. In the design of our holographic colour prints, the ability to control phase and amplitude on the level of individual pixels grants us the freedom to move pixels around (Fig. 3a) as long as

the phase is recalculated for any new pixel arrangement. As the total area allocated to each hologram is not fixed, we are also free to replace pixels of one hologram with pixels of another as long as each hologram still transmits enough light to give a reasonably high signal-to-noise ratio (Supplementary Note 2). Having the option to freely rearrange and replace pixels (spatial freedom) means that we are able to choose any arbitrary pixel arrangement for multiplexing, with only a few exceptions (Supplementary Note 6). In most cases, scrambling the pixel amplitudes and recalculating the phases (see Methods section for details of our design algorithm) does not greatly affect the fidelity of holographic projection when the number of pixels is sufficiently large (e.g. 480 × 480 pixels, as used here).

For simplicity, we first demonstrate the multiplexing of two holograms into a two-tone image, here a QR code. Although we designed these holograms for red and blue laser illumination, we note that there is no need for the colour filters used for multiplexing to be red and blue as long as their transmission amplitudes are mutually exclusive (orthogonal) at the design wavelengths. Because the requirement for wavelength selectivity constrains the spectra at only two points, there are a number of possible spectra and therefore colours that can be used. As such, we have some flexibility to choose colour filters with transmission spectra that best match both the desired image colours and the hologram design wavelengths ("spectral freedom"), or that otherwise achieve an optimal trade-off between these objectives.

Indeed, we found that red-and-blue QR codes provided very poor contrast and were difficult to read. To improve the visibility of our QR code for scanning under broadband white light illumination, we chose the colours yellow (with a high average transmittance of 48% over the range 450–650 nm, which contains most of the power of a typical white light source) and blue (low average transmittance of 22%) for the light and dark pixels, respectively. This choice of colours maximises the grayscale image contrast in white light while still retaining wavelength selectivity for the multiplexed holograms under monochromatic red and blue light (Fig. 3b).

Having selected suitable colours, exploiting the spatial degree of freedom in the multiplexed holograms allowed us to arrange the pixels into a print that shows a meaningful binary image (Fig. 3c), here a QR code that stores a 2620-bit message at error correction level H, which can be retrieved by scanning the image with a mobile phone or other reader. Notably, the holographic projection switches cleanly between the Chinese seal (Fig. 3d) and the Penny Black stamp (Fig. 3e) when toggling between red and blue laser illumination, despite both projections occupying the same region in space (Supplementary Fig. 7). Viewing at high magnification reveals the multiscale hierarchical structure of the print (Fig. 3f–h), in which pillar array colour filters of different dimensions and hologram phase plates of varying thickness are seamlessly integrated into the QR code pixels.

Finally, using the spectral degree of freedom, we relaxed the constraint on wavelength selectivity of the colour filters by introducing three additional colours (orange, yellow, and purple) into a three-colour multiplexed RGB (red, green, and blue) hologram and arranging the pixels to form a complex six-colour image (Fig. 4a). The additional colours were assigned to their closest matches within RGB: orange colour filters were placed over phase plates belonging to the red hologram, and purple colour filters over phase plates of the blue hologram (Fig. 4b). As yellow colour filters had poor selectivity between transmitting red light and green light (Fig. 4c), we opted not to store any hologram information in the phase plates under yellow filters—their high transmission at both wavelengths would cause the red projection to appear on the green channel as crosstalk and vice versa. However, if no information was stored in the yellow pixels, their

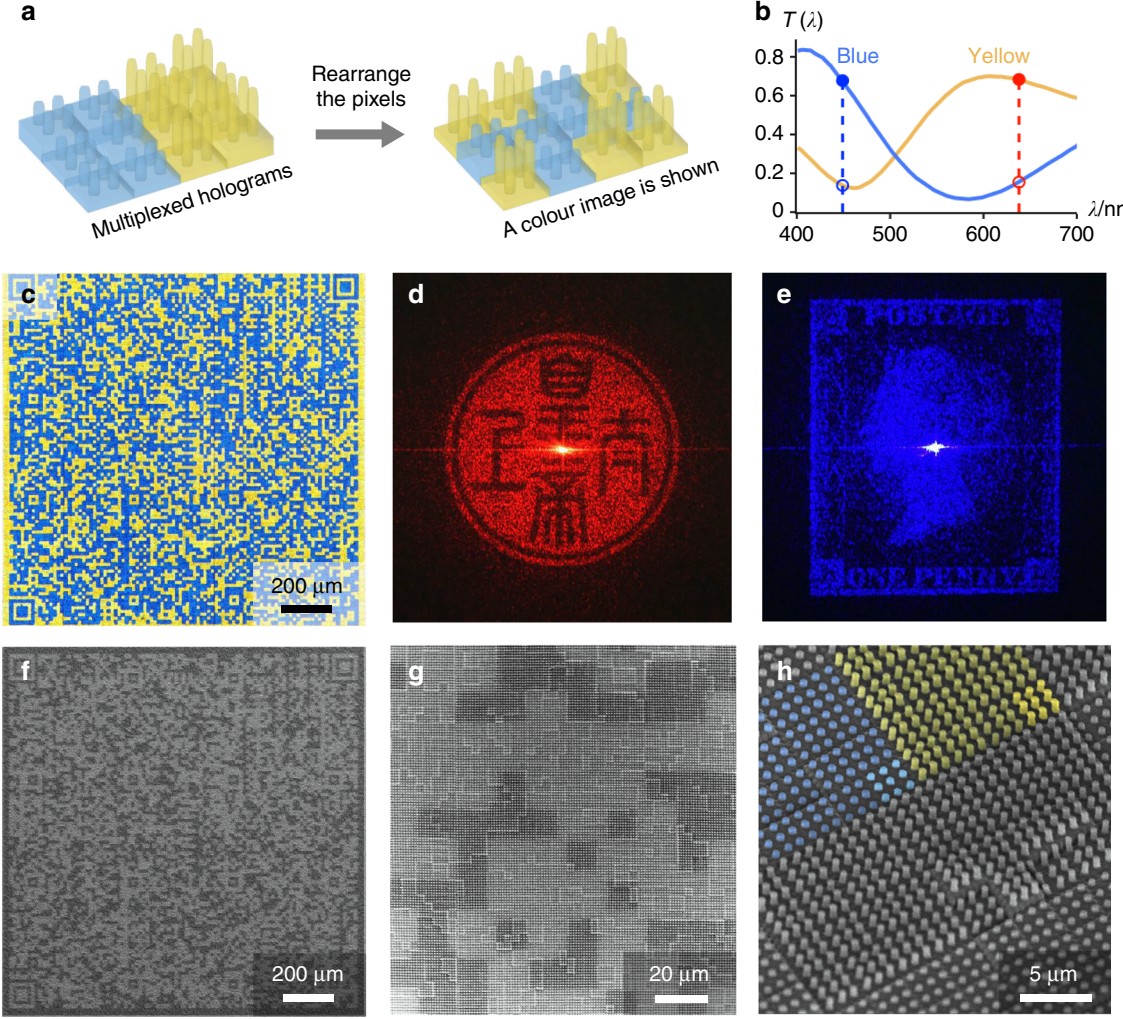

**Fig. 3** Proof-of-principle of a two-tone holographic colour print. **a** Design principle of the holographic colour print: The ability to choose the amplitudes of the pixels independent of their phase makes it possible to rearrange the pixels in a multiplexed hologram (spatial freedom) so that the pixel arrangement contains meaningful information as well, allowing a colour image to be shown on top of the holograms. As long as the pixel arrangement is taken into account when calculating the phase, the individual holographic projections can still be maintained. **b** Transmission spectra of the yellow and blue colour filters used, which have mutually exclusive (orthogonal) transmission at the wavelengths of interest (638 nm red and 449 nm blue). Optical characterisation of the print: **c** Transmission optical micrograph of a two-tone multiplexed hologram in which the pixels are arranged to form a 480 × 480 pixel colour QR code (1.44 mm square). The blue colour filters (blue hologram channel) selectively pass blue light but not red light, whereas the yellow colour filters (red hologram channel) selectively pass red light but not blue light. Holographic projections in transmission, photographed on a white wall in a darkened room: **d** the image of a Chinese seal is shown under 638 nm red laser illumination and **e** the image of a Penny Black stamp is shown under 449 nm blue laser illumination. The projection size scales with the projection distance, reaching an average size of ~10 cm at a distance of 1 m. **f-h** Scanning electron micrographs (SEMs) of the print at various scales. Each holographic colour pixel consists of a 3 × 3 pillar array colour filter on top of a 3 × 3 μm² phase plate, and each QR code pixel is a super-pixel comprising a 4 × 4 block of holographic colour pixels. In the close-up tilt-view SEM, a blue and a yellow QR code super-pixel are highlighted in false colour, and the bottom-right corner holographic colour pixel of each is further highlighted

constant phase would contribute to the transmitted zero-order (undiffracted) beam. Hence, we instead applied a random phase to diffuse the contribution from the yellow pixels into a uniform background.

In the final six-colour print, the high fidelity of holographic projections and remarkable lack of discernible crosstalk between them (Fig. 4d–f) demonstrates that it is possible to pattern complex and colourful images without sacrificing the quality of the multiplexed holograms in the same print. We note that the nature of the Perfume print allowed for the use of error diffusion dithering[23] in recolouring the image to obtain an optimal random pixel arrangement for high quality holograms (Supplementary Note 6). However, dithering could not be applied to the QR code print in Fig. 3 as it locally scrambles the colours and positions of

the pixels across the entire image, which would render the QR code impossible to scan. In order to accurately reproduce the QR code, we opted to retain its original (suboptimal) blocky pixel arrangement at the cost of slightly blurring the projections. The effects of different pixel arrangements on multiplexed holograms are compared in Supplementary Fig. 10.

## Discussion

A useful feature of our holographic colour prints is that it is easy to view both the colour image and the holographic projections without specialised equipment (Supplementary Note 3). The colour image can be captured by a handheld phone camera with a macro lens under narrow-beam white light illumination (Supplementary Fig. 3) such as that from a collimated flashlight.

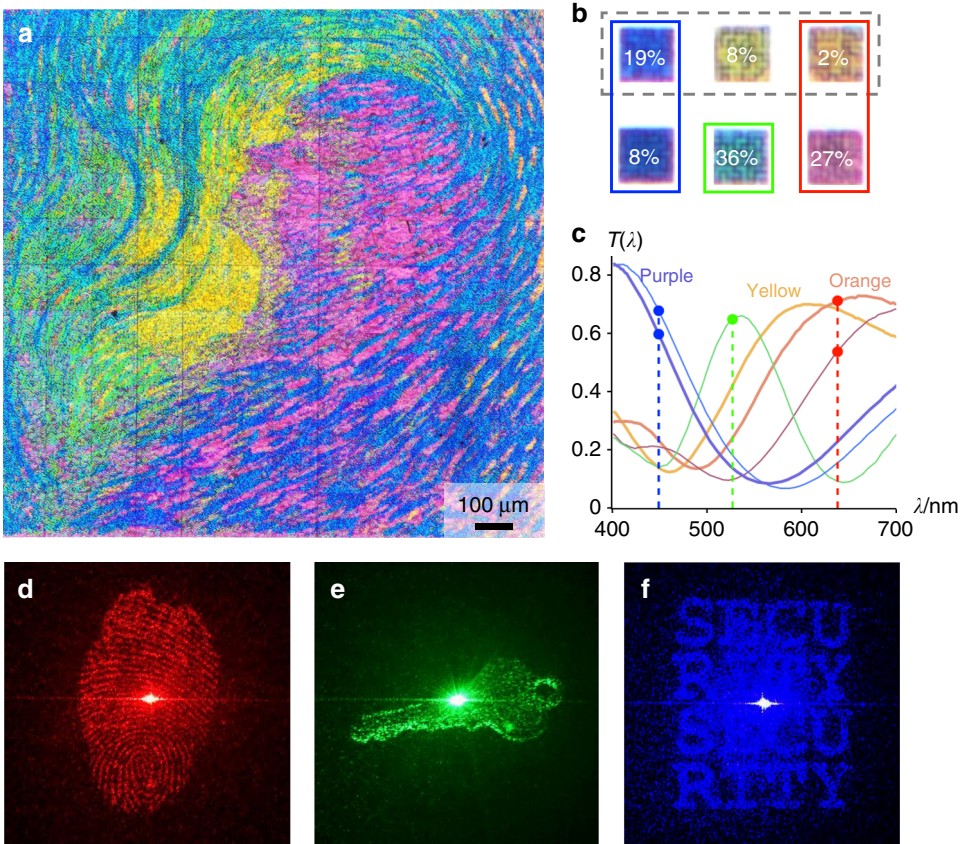

**Fig. 4** Enhanced optical security provided by a six-colour holographic colour print. **a** Transmission optical micrograph of the colour print, a 480 × 480 pixel (1.44 mm square) reproduction of Luigi Russolo's painting Perfume under which three holograms have been multiplexed. The ability to include colours that are less suitable for multiplexing the holograms (spectral freedom) allowed us to expand the usable colour palette to a total of six colours. **b** Optical micrographs of pillar arrays that produce the colours used in the print: added colours orange, yellow, purple (micrographs marked with a dashed box) and original colours red, green, and blue. Overlaid percentages on the micrographs of each pillar array denote the proportion of pixels in the print with that colour. Coloured boxes around the pillar micrographs indicate that they are used in the hologram channel of that colour. **c** Transmission spectra of pillar array colour filters used in the print: original colours in thin lines; added colours in thick lines. **d–f** Holographic projections of the print in transmission, photographed on a white wall in a darkened room: **d** a red thumbprint, **e** a green key, and **f** blue lettering that reads "SECURITY". Illumination sources were 638 nm red, 527 nm green, and 449 nm blue lasers, respectively. The projection size scales with the projection distance, reaching an average size of ~10 cm at a distance of 1 m. Original source images and simulation results of the colour image and holograms can be found in Supplementary Fig. 11

The holographic projections can be seen over a ~30° range of illumination angles (Supplementary Fig. 4), which is convenient for handheld viewing with a laser pointer. Due to the relatively high efficiency of the holograms (Supplementary Table 3), the projections from even a low power laser pointer are visible under normal room lighting (Supplementary Fig. 6). The overall experimental efficiency of our all-dielectric prints is as high as 10–20% for holographic laser projection and 30% for colour image transmission (Supplementary Note 5). Additionally, because the holographic projections appear on-axis, they are perfectly overlapped under collinear illumination and may potentially be used for full-colour projection (Supplementary Fig. 7).

We developed a monolithically integrated pixel that layers a structural colour filter over a phase plate to provide combined phase and amplitude control. Our design algorithm enabled us to create prints comprising large arrays of these pixels to simultaneously fulfil the objectives of hologram multiplexing and colour image formation. The holographic colour prints we fabricated are passive standalone devices capable of showing a colour image and multiple holographic projections under different illumination conditions. Because their phase and amplitude control is purely structural and the structures are made of a single material, our

prints can be completely described by a height map—in other words, information is stored only in their surface topography. Replicating a predesigned surface relief profile by nanoimprint lithography is thus a potential route towards manufacturing these prints. We anticipate that further research towards achieving pixel-level control of various properties of light may enable the development of other novel and practical optical security devices.

## Methods

**Designing a holographic colour print.** An iterative multi-objective MATLAB code was written to perform the colour image matching and phase calculation for each hologram channel (Fig. 5). Step 1 takes as input data a set of microscope images and transmission spectra collected from many different colour filters consisting of pillar arrays with varying dimensions (the spectra used were averaged from colour filters on blocks of several thicknesses, as described in Fig. 2). This data provides the colour as well as the transmittance (amplitude) of the pillar arrays at specific desired wavelengths. In Step 1, each pixel of the colour image to be patterned is colour-matched to the closest available colour in the dataset while balancing two considerations: the majority of the pixels in the image should have colours that are suitable for filtering RGB wavelengths, and the number of unique colours should be minimised to keep patterning time short. Once the colour filters are selected, the colour image is recoloured accordingly and the corresponding transmission spectra are used to generate a map of amplitudes (transmittance of each pixel at red, green, and blue wavelengths). This information is then fed into a modified Gerchberg–Saxton[24] algorithm (Steps 2–5) to iteratively optimise the

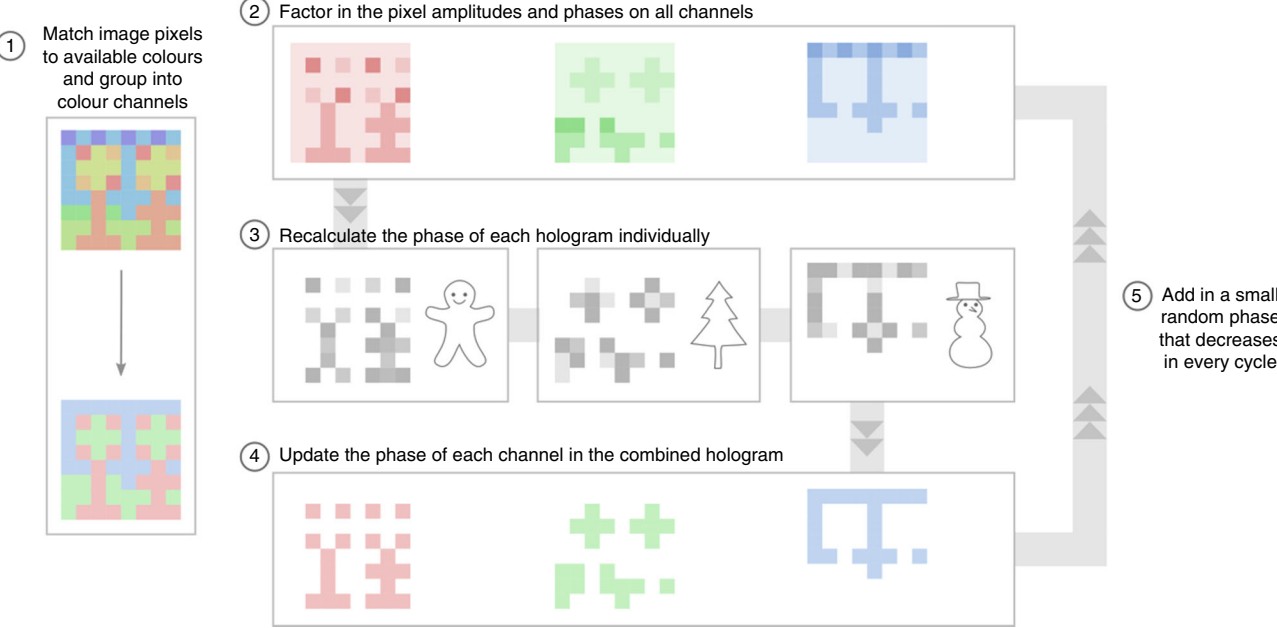

**Fig. 5** Design algorithm for holographic colour prints. Flowchart of the design algorithm for combining colour image formation and spatial multiplexing of holograms. The initial stage of the algorithm (Step 1) recolours the input colour image using a limited colour palette and then divides the colour pixels into several groups (channels) based on their suitability for filtering each hologram. After the assignment in Step 1, the main body of the algorithm (Steps 2–5) applies a modified Gerchberg–Saxton algorithm that takes into account the arrangement of pixels as well as their amplitudes (transmission spectra) and phases in order to iteratively re-optimise the phase of the holograms on each channel. Despite the imperfect selectivity of the pixel amplitudes (there is non-zero transmission at unwanted wavelengths, which results in crosstalk between channels), a satisfactory balance between the quality of the colour image and holographic projections can be achieved by using a sufficiently large number of pixels, which allows for both spatial (as exemplified by Fig. 3) and spectral (as in Fig. 4) degrees of freedom to be exploited

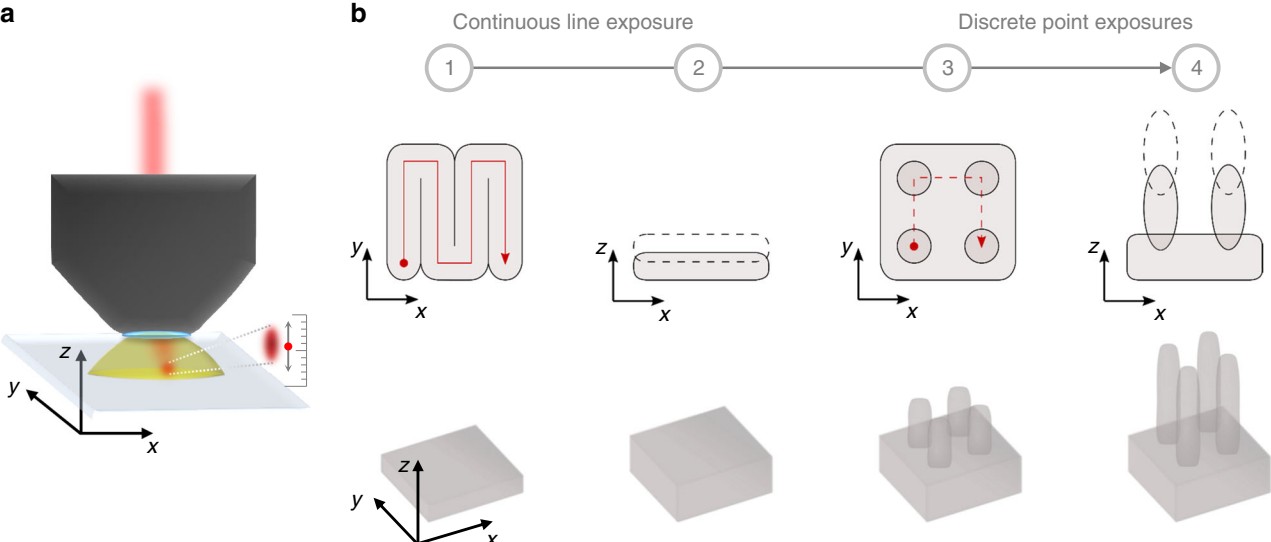

**Fig. 6** Fabrication process for holographic colour pixels. **a** In the direct laser writing exposure process, a 780 nm femtosecond pulsed IR laser beam is focused into a liquid puddle of negative-tone photoresist. At the focal point of the laser spot, the UV-sensitive photoresist is cross-linked by two-photon polymerisation and becomes solid. (Unexposed photoresist remains as a liquid and is later washed away during development.) The laser spot is translated in three dimensions (*x*, *y*, and *z*) according to the sequence in **b** to create complex structures. Resolution in the *z*-direction is not limited by the axially elongated shape of the point spread function as it is determined by the positioning accuracy of the laser spot. **b** Process flow for fabricating a holographic colour pixel using direct laser writing: (1) Blocks are created by rastering the laser spot to fill a square in the *xy*-plane with a continuous line exposure (hatching); (2) Step 1 is optionally repeated at higher *z*-positions to give thicker blocks (slicing); (3) Pillar arrays are created by point exposures, where the diameter is controlled by the exposure dose; (4) Step 3 is optionally repeated at higher *z*-positions to give taller pillars. Block thickness and pillar height are controlled in Steps 2 and 4 by overlapping multiple layers of exposures along the *z*-direction. A full holographic colour print is made by patterning an array of many pixels with different block thicknesses and pillar dimensions

phase of each element so as to best achieve three separate red, green, and blue grayscale holographic projections. We used a size of 480 × 480 pixels for our holographic colour prints, which could be computed in less than one minute by running our code on a modern laptop (Supplementary Methods).

**Fabricating a holographic colour pixel**. From the phase and amplitude maps created by the design algorithm in Fig. 5, a separate MATLAB code generated a blueprint of structures (phase plates and pillar array colour filters) with appropriate dimensions to achieve the desired phase and amplitude. This structural blueprint was finally converted into a set of instructions for controlling the laser writing sequence used in the fabrication process.

We fabricated holographic colour prints consisting of phase plates and colour filters in a single lithographic process by 3D direct laser writing on glass substrates. A femtosecond pulsed IR laser is focused by a high numerical aperture immersion microscope objective into the photoresist as a tight spot of submicron size (Fig. 6a). Two-photon absorption and polymerisation occur in the UV-sensitive photoresist at the focal point of the laser spot, which can be scanned laterally and shifted axially (refocused) relative to the photoresist/glass interface according to a predefined writing sequence to write a desired pattern consisting of points and lines (Fig. 6b). Cross-linking of the negative-tone photoresist along the laser exposure path creates the phase plates and colour filters as solid polymer structures on the glass.

The area to be patterned was split into a square grid of $120 \times 120\ \mu m^2$ write-fields based on the maximum undistorted field of view of the microscope objective. The write-fields were written sequentially and stitched together by successive translations of the stage on which the substrate was mounted. In each write-field, we grouped blocks of the same thickness and performed the writing sequence in Fig. 6b for each group in ascending order of thickness. Within each group of blocks, all blocks were patterned before their pillars were patterned. We chose to group the blocks by thickness instead of spatial coordinate to minimise patterning time, as refocusing in the $z$-direction is much slower than lateral scanning in the $xy$-plane. The total writing time for a 1.44 mm square print (480 × 480 array of 3 μm pixels) was 6–8 h.

**Code availability**. The code used in this work can be made available upon request.

## Data availability
Transmittance values of the colour filters, efficiency measurements of the prints, and source images used for the holographic projections and the Perfume colour print are provided in the Supplementary Information.

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

## Acknowledgements
We acknowledge funding support from SUTD Digital Manufacturing and Design Centre Grant Nos. RGDM1530302 and RGDM1830303, and Singapore National Research Foundation Grant No. NRF-CRP001-021. We thank Junjie Kang for technical assistance in assembling a setup for collinear RGB illumination, Eric Tan for photography of the holograms, Shuang Zhang for introducing us to the Gerchberg–Saxton algorithm, and Cheng-Wei Qiu for useful discussions.

## Author contributions
K.T.P.L. devised the concept, designed the code, performed experiments, and wrote the manuscript. H.L. and Y.L. performed experiments. J.K.W.Y. edited the manuscript and supervised the research.

## Additional information

**Competing interests:** The authors declare no competing interests.

