## [Peer Review File · Nature Communications]

Reviewers' Comments:

Reviewer #1:

Remarks to the Author:

1. The authors state "the wide colour range enables us to select suitable colours to reproduce the colour image under white light illumination", how do authors support the point of wide colour range? There is not enough data to support this point. For the structural colour device, both colour saturation and colour gamut are very important indicators. Furthermore, there are obvious fabrication errors, as shown in fig. 2c, so is there a big difference between the designed colour and the measured colour? I strongly suggest the authors to discuss and analyze these in the manuscript.
2. Furthermore, there are some descriptions that overstate performance. For example, the authors emphasized many times in the manuscript that their designed structures have good wavelength selectivity. In fact, the signal-to-noise ratio is not high, only about 3:1, which in return leads to crosstalk between different holography projections, as shown in fig. 3c and fig. 4c. Especially for fig. 4c, the crosstalk has made it hard to distinguish the letters in the blue projection. In fact, the projections displayed in Fig. 4 are separated in space, which are not sufficient to realize true colorful holograms. The authors should analyze the results and evaluate their work more objectively.
3. It is not clear how such device can be used in optical security applications, since once light illuminates, the patterns will be generated. Regarding encryption and decryption, the key should be complex enough to maintain the security of information.
4. Please give the parameters of the structures corresponding to three added colours in fig. 4b.
5. For the phase control, why are there different discrete thickness levels for different colors? Is that for a special purpose?
6. Please give the original structural color images and calculated holography images in the manuscript in order to enable readers to directly evaluate the performance of devices.

Reviewer #2:

Remarks to the Author:

The manuscript by Yang et al. reports a holographic color print by combined phase and amplitude control. With the spatially multiplexed pixel design that combines phase plates and structural color filters, they can achieve simultaneously color image formation and hologram multiplexing under white light and coherent laser illumination, respectively. The entire print is fabricated in a single lithographic process using a femtosecond 3D direct laser writer. Such method can find applications in document security. The paper is well written and logically organized. The topic is timely and I think broad enough to be suitable for Nature Communications. However, some aspects of the work must be addressed before I can make a final decision for the recommendation of publication in Nature Communications.

1. The concept of the holographic color print is composed of a color filter layer and a bottom layer as a phase plate. The authors claim that the two layers can be independently controlled, but actually the two layers can interact simultaneously to the incident light. That is, the interaction of light by the integration of the two layers can cause extra amplitude and phase noise by comparison to an independent design. Here, the authors need to provide a more detailed analysis of the design procedures for the two layers and the interaction of the two layers?
2. The authors should provide all the geometry parameters of the dielectric color filter and phase plates in their manuscript. Otherwise, the work cannot be reproduced and checked for consistency.
3. The authors mention that the holographic projection can be achieved over a wide range of incident angles. Why is this the case? In general, the phase and amplitude modulation are sensitive to incident angle. Here, a discussion and some experimental results of the angular dependence needs to be added to the manuscript and to support such statement.

4. Indeed, such method may suffer from cross-talk between different incident wavelengths. In order to improve the image qualities, the authors use a dithering algorithm to provide a white noise spectrum. To fully understand such method the authors need to provide more details of such a method? Furthermore, in the discussion they mention "Fig.3 should be understood as a worst case scenario for the performance." Here, I would expect some information about how the authors can further improve the imaging quality?
5. To put the work into context with other concepts the authors should provide values for the experimental efficiency for white light illumination and the coherent holographic projection?

Reviewer #3:

Remarks to the Author:

In this manuscript, the authors present a technique to design, fabricate and demonstrate multiplexed color holograms within a single color image. In their demonstration, the holographic color print appears as a color image when illuminated with white light, whereas it projects three different holograms when illuminated by red, green or blue laser beams. The field of holography is a popular field and has implications in security systems. This work incorporates the concept of multiplexed holograms and color filters in a single device and has sufficient novelty in its field of research. This work may be of interest to the general optics community and specifically to researchers working on holography.

The manuscript is well-written, well organized and the results and discussions are presented in a coherent and orderly manner.

However, I would suggest that this manuscript be revised and the following issues be addressed in a revised version before the work can be considered for publication:

1. What is the size of the color print as shown in Fig. 3(c)? From the scale bar, it appears to be an overall size of $\sim 1.2 \text{ mm} \times 1.2 \text{ mm}$. It is helpful to mention this size in the text or the figure captions.

2. One of the issues that I find with this work is that the size of the color print is very small. At $\sim 1 \text{ sq. mm}$, it is difficult or impossible to see any detail with the naked eye, if an optical microscope is not used. It seems as if this is counter to the idea that the color print can be viewed easily. Have the authors considered this fact? How large can these prints be made?

3. What is the relation between the size of the color print region and the size of the projected hologram? I believe this is an important point to consider and seems that for a fairly large projection ($\sim 10 \text{ cm}$ at 1 m distance), the size of the color print has to be relatively small, as demonstrated here. Can the authors add a section that discusses this?

4. It also seems that from a nanofabrication perspective, that the fabrication using two photon lithography would become time consuming for writing structures over large areas. What was the total lithography write times (approximate) for the two devices (prints) demonstrated?

5. What is the computational power required to design these prints using the algorithm described? It is an iterative process, as is evident from the description. The authors should mention the computational power (processor) and total time taken to converge to a useful design.

Overall, the quality of the work is high. However, the authors make the claim in the introduction and discussions of the usability of this technique to security systems, where, even though the proof-of-concept results are interesting, the applicability of the method to large scale would have to be thoughtfully considered given the issues that I have raised above.

In view of the above-mentioned issues, my decision is for the manuscript to be revised and the issues addressed in a response letter.

Reviewers' comments:

Reviewer #1 (Remarks to the Author):

1. The authors state "the wide color range enables us to select suitable colors to reproduce the color image under white light illumination", how do authors support the point of wide color range? There is not enough data to support this point. For the structural color device, both color saturation and color gamut are very important indicators. Furthermore, there are obvious fabrication errors, as shown in fig. 2c, so is there a big difference between the designed color and the measured color? I strongly suggest the authors to discuss and analyze these in the manuscript.

Color range:

In addition to showing the set of colors accessible using the pillar-on-block geometry in the transmission optical micrograph in Fig. S1, we have now included the CIE color space plots of the corresponding transmission spectra in Fig. S2. We measured an area coverage of 53% relative to the sRGB color gamut.

Fabrication errors:

Indeed, the expected (designed) color and actual (measured) color can sometimes differ. However, when these errors occur it is due to variations in the fabrication process rather than imperfections in the fabrication. As our colors are chosen empirically by an iterative process of fabricating and characterizing test structures, exploring the parameter space until we arrive at suitable structures, common imperfections (such as the bending of the pillars seen in Fig. 2c) are naturally taken into account in our selection process. Thus there is no strict requirement for the pillars to be perfectly straight in order for us to obtain a desired color, only for their fabrication to be reasonably consistent.

2. Furthermore, there are some descriptions that overstate performance. For example, the authors emphasized many times in the manuscript that their designed structures have good wavelength selectivity. In fact, the signal-to-noise ratio is not high, only about 3:1, which in return leads to crosstalk between different holography projections, as shown in fig. 3c and fig. 4c. Especially for fig. 4c, the crosstalk has made it hard to distinguish the letters in the blue projection. In fact, the projections displayed in Fig. 4 are separated in space, which are not sufficient to realize true colorful holograms. The authors should analyze the results and evaluate their work more objectively.

Wavelength selectivity:

We have listed the transmittance values (Table S1). The average "high" transmittance is 62% while the average "low" transmittance is 15%. We also calculated the wavelength selectivity (Table S2) of our red, green, and blue color filters, obtaining values between 3.2 and 6.6 (average of 4.6). So in fact 3:1 is the worst-case selectivity, and is mitigated in our prints by

adjusting the fraction of pixels allocated to each channel as described in the new section in the Supplementary Information on “Balancing wavelength selectivities for hologram multiplexing”.

Crosstalk:

We added more details on how we balance the signal-to-noise ratio on the three channels in a multiplexed hologram by adjusting the relative area allocation of pixels, which allows us to achieve an optimal outcome with minimized crosstalk. This information can be found in the section on “Balancing wavelength selectivities for hologram multiplexing” in the Supplementary Information.

Projections separated in space:

The referee might have misunderstood what we have demonstrated here. In both Figs. 3 and 4, the holographic projections are not actually separated in space but overlapped on-axis. Because of this overlap, we switch on the lasers one at a time so that the individual projections are clearly seen. To emphasize this point, we have added Fig. S7 where the overlap of the projections is seen with two collinear lasers switched on simultaneously. We observe that the crosstalk is in fact remarkably low for on-axis overlapped projections.

3. It is not clear how such device can be used in optical security applications, since once light illuminates, the patterns will be generated. Regarding encryption and decryption, the key should be complex enough to maintain the security of information.

We would like to clarify that we refer specifically to optical document security in the text (i.e. authentication features in banknotes or documents), and have not claimed optical encryption or information storage as an application of our work. Thus there is no need for keys, although it would in principle be possible to encode the information using a phase mask as a key. This has been further clarified in the text.

4. Please give the parameters of the structures corresponding to three added colors in fig. 4b.

Parameters of all structures used are now provided in the Supplementary Information.

5. For the phase control, why are there different discrete thickness levels for different colors? Is that for a special purpose?

For each wavelength λ , we choose the maximum useful number of phase steps (thickness levels) based on the thickness $t = \lambda/(n - 1)$ required to achieve 2π phase modulation and the thickness accuracy dt . Limited by a constant accuracy of $dt \sim 100$ nm, the maximum useful number of thickness levels $N = t/dt \propto \lambda/dt$ decreases at shorter wavelengths.

It follows that the actual thickness levels themselves must be different – it is mathematically impossible for different wavelengths to share the same thickness levels because the interval endpoints and step sizes are different for each wavelength (color) when the numbers of thickness levels are coprime.

6. Please give the original structural color images and calculated holography images in the manuscript in order to enable readers to directly evaluate the performance of devices.

Source images and calculated images are now provided in Fig. S11.

Reviewer #2 (Remarks to the Author):

The manuscript by Yang et al. reports a holographic color print by combined phase and amplitude control. With the spatially multiplexed pixel design that combines phase plates and structural color filters, they can achieve simultaneously color image formation and hologram multiplexing under white light and coherent laser illumination, respectively. The entire print is fabricated in a single lithographic process using a femtosecond 3D direct laser writer. Such method can find applications in document security. The paper is well written and logically organized. The topic is timely and I think broad enough to be suitable for Nature Communications. However, some aspects of the work must be addressed before I can make a final decision for the recommendation of publication in Nature Communications.

1. The concept of the holographic color print is composed of a color filter layer and a bottom layer as a phase plate. The authors claim that the two layers can be independently controlled, but actually the two layers can interact simultaneously to the incident light. That is, the interaction of light by the integration of the two layers can cause extra amplitude and phase noise by comparison to an independent design. Here, the authors need to provide a more detailed analysis of the design procedures for the two layers and the interaction of the two layers?

Design procedures for the color filter layer and phase plate layer are described in the Methods section and Fig. 5 of the paper. Briefly, the input color image is recolored based on the available palette of color filters and the recolored image is then used to set the map of amplitudes for the subsequent hologram phase (phase plate thickness) optimization.

We found a range of phase plate thicknesses (0.6–1.8 μm) that minimally affects the colors of the filters (Fig. S1), such that the pixel amplitude is mostly unaffected by its phase. The comparable gamut achieved by filters on phase plates of random heights, and filters directly patterned on glass supports this claim.

However, to determine whether the pixel phase is affected by its amplitude is more difficult, i.e. whether changing the pillar dimensions in the color filter layer could affect the holograms in the phase plate layer below it. To this end, we developed an experiment as follows: We fabricated a test sample of checkerboard gratings made of pillar arrays of similar transmittance at the R, G, or B wavelengths, but different color and heights (Fig. S8). From power measurements of the first order diffraction spot, we observed that the pillar array gratings were an order of magnitude less diffractive as compared to gratings made of phase plate blocks (Fig. S9), which indicates that they add only a small phase to the integrated pixels. As such, we conclude that the interaction of the two layers (phase influencing amplitude, and amplitude influencing phase) is relatively weak. We note that it would be quite straightforward to correct for the phase of the color filters if this additional phase imparted by the color filter was known, i.e. by pre-compensating with the thickness of the phase plate blocks to keep the phase at a constant desired value.

We have added the above discussion to the Supplementary Information.

2. The authors should provide all the geometry parameters of the dielectric color filter and phase plates in their manuscript. Otherwise, the work cannot be reproduced and checked for consistency.

Parameters of all structures used are now provided in the Supplementary Information.

3. The authors mention that the holographic projection can be achieved over a wide range of incident angles. Why is this the case? In general, the phase and amplitude modulation are sensitive to incident angle. Here, a discussion and some experimental results of the angular dependence needs to be added to the manuscript and to support such statement.

We have performed an extensive investigation of the angular and distance dependence of the holographic projections, which has been added as a section in the Supplementary Information. Experimental results show a 30° tolerance range for the incident angle of illumination (Fig. S4).

In our holograms, the thickness of the phase plate blocks is smaller than their lateral size. As such, the holograms are too thin to accommodate even their smallest grating period (twice the block size) along the vertical direction and thus cannot act as thick, volume Bragg gratings (which are highly angle selective). This “thin hologram” regime is characterized by holograms with relatively large angle independence and having multiple diffraction orders, both of which are observed in many Fraunhofer-type computer-generated holograms. We now discuss this in the text of the Supplementary Information.

4. Indeed, such method may suffer from cross-talk between different incident wavelengths. In order to improve the image qualities, the authors use a dithering algorithm to provide a white noise spectrum. To fully understand such method the authors need to provide more details of such a method? Furthermore, in the discussion they mention “Fig.3 should be understood as a worst case scenario for the performance.” Here, I would expect some information about how the authors can further improve the imaging quality?

Dithering algorithm:

We have added an explanation of the common Floyd-Steinberg error diffusion dithering algorithm as implemented as a MATLAB built-in function in the Supplementary Information. The discussion on “worst case scenario” has been revised. The image quality of the holographic projections in Fig. 3 cannot be improved further except perhaps by increasing the number of pixels used in the print to gain more degrees of freedom. The fundamental limitation is that we cannot afford to let the color image pixels be scrambled since it is a QR code and intended for scanning, but this also means that error diffusion dithering cannot be used to improve the appearance of the holographic projections.

5. To put the work into context with other concepts the authors should provide values for the experimental efficiency for white light illumination and the coherent holographic projection?

A detailed investigation of the experimental efficiency for holographic projection has been performed on the QR code print. By taking measurements with a power meter, we found that the filter transmission efficiency was ~34%, as expected for pixels with 68% transmittance covering an area fraction of ~50%. The hologram diffraction efficiency was 72% (blue) and 47% (red). The overall efficiency was determined to be 21% (blue) and 14% (red).

In the *Perfume* print, which has a representative mix of colors, the average transmission efficiency for white light illumination (450-650 nm) is 30%, which we calculate by taking a weighted average of the transmittance values (weighted by the area fraction in the print occupied by each color). We have now added this information as a new section in the Supplementary Information (Table S3).

Reviewer #3 (Remarks to the Author):

In this manuscript, the authors present a technique to design, fabricate and demonstrate multiplexed color holograms within a single color image. In their demonstration, the holographic color print appears as a color image when illuminated with white light, whereas it projects three different holograms when illuminated by red, green or blue laser beams. The field of holography is a popular field and has implications in security systems. This work

incorporates the concept of multiplexed holograms and color filters in a single device and has sufficient novelty in its field of research. This work may be of interest to the general optics community and specifically to researchers working on holography.

The manuscript is well-written, well organized and the results and discussions are presented in a coherent and orderly manner.

However, I would suggest that this manuscript be revised and the following issues be addressed in a revised version before the work can be considered for publication:

1. What is the size of the color print as shown in Fig. 3(c)? From the scale bar, it appears to be an overall size of ~ 1.2 mm x 1.2 mm. It is helpful to mention this size in the text or the figure captions.

The size of the color print is 1.44 mm x 1.44 mm. We have now added this information in the text and the figure captions.

2. One of the issues that I find with this work is that the size of the color print is very small. At ~ 1 sq. mm, it is difficult or impossible to see any detail with the naked eye, if an optical microscope is not used. It seems as if this is counter to the idea that the color print can be viewed easily. Have the authors considered this fact?

We have now added a section on “Practical applicability of holographic colour prints” in the Supplementary Information in which we investigate in detail the ease of viewing of the colour image and holographic projections.

In this section, Fig. S3 shows that the color print can be photographed with a smartphone camera mounted with a low cost macro lens. We believe it is not necessary for the prints to be viewed directly by eye as long as they can be seen with the aid of commonly or cheaply available tools. This is also the case for many existing optical document security features, such as hidden images rendered in various forms of invisible ink (e.g. UV, IR), which provide a level of covert security.

How large can these prints be made?

The only physical limitation on the largest prints is the size of the substrate that can be mounted and the maximum stage travel of the Nanoscribe Photonic Professional GT direct laser writer that we used. The current tool is specified to pattern up to ~ 10 cm square area. However, for the purpose of demonstrating the key concepts in this manuscript, we have refrained from large-area prints, which could be a topic for future work.

3. What is the relation between the size of the color print region and the size of the projected hologram? I believe this is an important point to consider and seems that for a fairly large projection (~ 10 cm at 1 m distance), the size of the color print has to be relatively small, as demonstrated here. Can the authors add a section that discusses this?

There is no direct relation between the size of the color print and the size of the projected hologram. A large print with the same pixel density would exhibit the same angular extent. It is the size of the individual pixels that controls the angular extent of the projected hologram according to an inverse relationship (the ratio of wavelength to pixel size is equal to the ratio of the projection size to projection distance). We now discuss this inverse relationship in Fig. S5, comparing the diffraction angles of the holograms and the color filters.

The size of the color print is the product of the pixel size and the number of pixels. Hence if the number of pixels is kept the same but the print is scaled up in size, the angular extent of the projected hologram will decrease. However, one can maintain the angular extent of the hologram by increasing the total number of pixels to keep the pixel density constant. Doing so also increases the level of detail in the hologram as the information storage capacity of the hologram and resolution of the projection (the space-bandwidth product) scale with the number of pixels. In our work, we have limited our prints to a few millimeters as that is the size of the laser spot used in illuminating the samples.

4. It also seems that from a nanofabrication perspective, that the fabrication using two photon lithography would become time consuming for writing structures over large areas. What was the total lithography write times (approximate) for the two devices (prints) demonstrated?

The total write time was 6-8 hours for each print. (We have now added this detail in the text.) As this is a proof of concept demonstration, we believe that the write time is reasonable and comparable to that of large or complex prints performed with electron-beam lithography. In fact, our prints are much larger and more complex than is practical to create with conventional electron-beam lithography. Note also that the 3D nature of the print precludes the use of any other existing equipment. Still higher speed could be achieved through improvements in instrumentation -- scanning the laser spot and performing stage movements faster while retaining sufficient accuracy, and using higher power lasers and more sensitive photoresists to decrease exposure time. From a manufacturing perspective, it may also be possible to scale up using nanoimprint lithography as all of the information in the prints is stored in the surface topography of the structure and is amenable to replication.

5. What is the computational power required to design these prints using the algorithm described? It is an iterative process, as is evident from the description. The authors should

mention the computational power (processor) and total time taken to converge to a useful design.

We mention in the paper that the time taken for the algorithm to run is under a minute. We have now added details of the processor used (Intel i5-7300U 2.60 GHz, with 8 GB RAM).

Overall, the quality of the work is high. However, the authors make the claim in the introduction and discussions of the usability of this technique to security systems, where, even though the proof-of-concept results are interesting, the applicability of the method to large scale would have to be thoughtfully considered given the issues that I have raised above.

In view of the above-mentioned issues, my decision is for the manuscript to be revised and the issues addressed in a response letter.

Reviewers' Comments:

Reviewer #1:

Remarks to the Author:

The revision is reasonable and I recommend its publication now.

Reviewer #2:

Remarks to the Author:

The authors have thoroughly answered all the questions, clarified some confusing points of the previous version, and offered sufficient revisions in the manuscript and the supplementary material. Due to the novel aspects and high interest of the topic, I prefer to recommend its publication in Nature Communications.

Reviewer #3:

None